# Experiences of Forest Healing Instructors Who Met Cancer Patients in Forest Healing Programs: FGI Research

**DOI:** 10.3390/ijerph20054468

**Published:** 2023-03-02

**Authors:** Eun Young Park, Min Kyung Song, Mi Young An

**Affiliations:** 1College of Nursing, Gachon University, Incheon 21936, Republic of Korea; 2Department of Nursing, College of Medicine, University of Ulsan, Ulsan 44610, Republic of Korea; 3Department of Nursing, Yeoju Institute of Technology, Yeoju 12652, Republic of Korea

**Keywords:** integrated care, cancer survivor, forest healing program, forest healing instructors

## Abstract

Background: Demand for urban forest programs for the healing of cancer survivors is increasing. To develop a forest healing program for the integrated care of cancer patients, it is necessary to analyze the experiences of forest healing instructors who have conducted programs for cancer patients. Methods: This qualitative study applied focus group interviews (FGIs; four interviews with sixteen participants) to describe and understand the experiences of forest healing instructors who run forest healing programs for cancer patients. Results: Four themes were identified: “prepared encounters and unexpected encounters,” “yearning for healing,” “people who need special care,” and “things to prepare for cancer patient programs.” Conclusion: Forest healing instructors had difficulty facilitating programs for cancer patients owing to prejudice and a lack of knowledge about the characteristics of cancer patients. Moreover, differentiated programs and places that align with the specific needs of cancer patients are needed. It is necessary to develop an integrated care forest healing program for cancer patients and educate forest healing instructors about the needs of cancer patients.

## 1. Introduction

In 2020, approximately 19.3 million people worldwide were newly diagnosed with cancer and nearly 10 million died [1,2]. Cancer has been the leading cause of death in Korea over the past decade [3] and is considered a major health problem. In 2019, 254,718 people in Korea were newly diagnosed with cancer, representing an increase of 3.6% from 245,874 in 2018 [4]. The number of newly diagnosed cancers and the prevalence of cancer continue to increase [5]. Survival rates of cancer patients are increasing owing to early detection and development of cancer treatment technologies [6]. In Korea, the 5-year relative survival rate of cancer patients in the years 2015–2019 was 70.7%, an improvement of 5.2% from 65.5% in 2006–2010 [4]. Consequently, interest not only in the treatment of cancer patients but also in health management is increasing. Cancer patients require management of recurrence and complications, as well as management of physical symptoms and psychosocial problems [6]. Accordingly, many studies emphasize the need for an integrated multidisciplinary approach [7].

Interest in forests as places for rest and healing is growing along with improvements in living standards and increasing life expectancy [8]. “Forest healing” refers to immune-strengthening and health-promoting activities that utilize a variety of elements of the forest, including scents and scenic views [9]. Forest healing programs are reported to be effective for health problems such as hypertension, stress, depression, and anxiety [10,11]. A systematic review of the effects on adult patients with various diseases indicated positive effects not only on physiological indicators, such as blood pressure, natural killer cells, and perforin, but also on psychosocial indicators [12]. Owing to these positive effects, people affected with various types of diseases are visiting urban forests for rest and healing, and the demand continues to increase [13]. Forest healing in cancer patients reportedly reduces the level of cortisol, a stress hormone [14], and improves sleep quality [15]. It has also been reported to improve cancer-related fatigue and emotional fluctuations [16]. Cancer patients actively utilize natural products and implement lifestyle modifications along with existing cancer treatments to improve their health and quality of life throughout the treatment process [17]. Forest healing programs include increased physical activities and the utilization of nature. Forest therapy could be a beneficial and complementary approach for cancer patients to manage their symptoms and improve their overall well-being.

In Korea, forest healing instructors who plan and develop customized forest-based programs for individuals are trained to effectively facilitate forest healing activities. The efficacy of forest healing programs is affected by the expertise and competency of the individual instructors who develop and guide the program that can influence the cognition, emotion, and behavior of forest healing program participants. As they can directly affect the efficacy of the healing programs, the role of forest healing instructors is considered to be extremely important [18].

Although interest in forest healing programs is growing, related research is mostly focused on the participants, and there are very few studies on the forest healing instructors who operate the programs. A recent qualitative study on forest healing instructors’ overall program operation experience highlighted difficulties due to the various ages of the participants, and tailoring of the contents of the program to the characteristics of the participants’ diseases was insufficient [19]. It is important to ensure that forest healing programs are tailored to the specific needs of cancer patients. By developing a forest healing program for cancer patients, it is possible to improve their physical and emotional well-being, enhance their quality of life, and provide them with a valuable resource for managing the challenges of cancer treatment and recovery. As the development of customized programs centered on participants is a continuous process, it is necessary to take the experiences of forest healing instructors who have encountered cancer patients during forest healing programs into account.

Therefore, this study aimed to understand the perspectives of forest healing program instructors, who had encountered cancer patients during forest healing programs, on cancer patients and the forest healing program process for cancer patients through their experiences. The research results can be used to develop and improve forest healing programs for cancer patients and to provide training and professional development for forest healing instructors. In addition, they can contribute to the field of integrative medicine and inform healthcare professionals and policymakers about the potential benefits of forest healing programs for cancer patients.

## 2. Methods

### 2.1. Study Design

This qualitative study applied focus group interviews (FGIs) to describe and analyze the experiences of forest healing instructors who facilitate forest healing programs for cancer patients.

### 2.2. Participants and Recruitment

The inclusion criteria for participants in this study were as follows: those with a forest healing instructor license; those who had experience in facilitating forest healing programs; those who had encountered cancer patients in forest healing programs; and those who voluntarily agreed to participate in this study. Participants were recruited through email communications, online posters, and social media posts by the Forest Healing Instructor Association.

### 2.3. Definition of Terms

Forest healing: Forest healing is an activity that utilizes various environmental factors that exist in the forest to enhance the body’s immunity and restore physical and mental health. Forest healing is not a cure for disease but a healing activity that helps to maintain health and enhance immunity [20].Forest healing instructor: Forest healing instructors are nationally certified experts who plan and develop customized forest healing programs for specific target groups using forests, including healing forests and natural recreation forests, to support efficient forest healing activities. The forest healing instructor license is a national license issued by the head of the Korea Forest Service and is divided into advanced instructor and instructor according to qualification standards [20].

### 2.4. Data Collection

Data were collected from four FGIs conducted between August 7 and August 29, 2021. Each focus group consisted of 3–5 participants with experience in conducting forest healing programs for cancer patients. The interview duration ranged from 2 to 2.5 h. The FGI used semi-structured questions, including introductory, transition, and key questions [21]. In the opening session, the participants were asked about their careers and current work. Following this, an opening question was asked to elicit important information. The main interview questions included: “Do you remember any participants who participated in the forest healing program?”, “How was your first meeting with a cancer patient during the program?”, “What do you think about cancer patients?”, and “What is the most memorable activity that patients with cancer participated in?” Data were collected using follow-up questions until data saturation. Three researchers participated as interviewers and field observers in each FGI. To ensure clarity and accuracy of the transcription, two researchers who participated as on-site observers recorded the interviews with the consent of the participants and wrote on-site notes.

### 2.5. Data Analysis

All interviews were recorded on-site and transcribed by trained researchers. Data were analyzed according to the qualitative thematic analysis method proposed by Braun and Clarke [22]. First, the transcribed data were read repeatedly to understand their overall meaning. Second, open coding was used for analysis. The research team read the transcript and marked phrases and sentences containing keywords or meanings on the transcript to classify and code the units of analysis to create a list. Third, to identify themes, similarities and differences in the coding list were compared and classified into similar semantic units. Fourth, the extracted semantic units were reviewed again to conduct a detailed analysis, and the meaning of each subject was confirmed by reviewing the original data. Fifth, through further discussion within the research team, the final theme representing the essence of the experience was defined, named, and contextualized. The transcripts and on-site notes of the participant interviews were reflected in the data analysis.

### 2.6. Rigor

To ensure the rigor of the study, we applied Sandelowski’s reliability, suitability, auditability, and verifiability criteria [23]. We ensured the accuracy of the transcribed material compared to the recording for reliability. We also checked the adequacy of citations in support of data-derived themes for suitability. Participants’ narratives were explained and interpreted based on the derived themes. For auditability, we read quotations describing the topic, selected the one that best embodied the topic, and explained the analysis procedure. We shared and discussed the experiences and opinions obtained from the data collection and analysis to maximize verifiability and minimize bias and attempted to maintain neutrality through reflective journaling throughout the research process.

### 2.7. Ethical Considerations

This study was approved by the Institutional Ethics Review Board (IRB: 1044396-202103-HR-055-01). The research team explained the purpose and process of the study and obtained written consent from the participants to record the FGI. It was also explained that the data would be used for research purposes only and that recordings and manuscripts would be stored on a password-protected computer to ensure maximum safety.

## 3. Results

### 3.1. General Characteristics of Participants

A total of 16 participants were included in the study. They were divided into four teams and completed FGIs. The average age of the participants was 58.44 years, 75.0% of the participants were women, and 75.0% received graduate school or higher education. The majority of participants were majors in forest science (37.5%) and medicine, nursing, and public health (37.5%), and 25.0% were majors in other areas. Regarding their qualifications, 75.0% of the participants had forest healing instructor licenses, and 25.0% had both instructor and advanced instructor forest healing licenses (25.0%). Among the forest healing instructors, 56.3% qualified as forest interpreters, 87.5% were currently active forest healing instructors, and 31.3% were active in urban parks (Table 1).

### 3.2. Qualitative Thematic Analysis

Based on the experiences of forest healing program instructors, four themes and 11 sub-topics were analyzed (Table 2). These themes were derived from 340 semantic phrases initially derived from four FGI datasets.

#### 3.2.1. Prepared Encounters and Unexpected Encounters

This theme concerns the experiences of forest healing instructors who first met cancer patients in forest healing programs. Many participants said that they were bewildered when they discovered that there were cancer patients among the program participants.

##### Request for a Special Program for Cancer Patients

Participants who ran a cancer integration support program at a local public health center and a program for cancer survivors in collaboration with a nearby forest healing center said that they were impressed by their lively and bright appearance, contrary to their preconceived notions about cancer patients.

“They (cancer survivors) are much more positive than the general person we met before at forest. Just do what you prepared and there will be no problem”.(FGI B-3)

In contrast, participants who ran an advanced cancer patient program experienced difficulties in running the program for each session because cancer patients were often unable to participate continuously owing to changes in their health status. Based on this experience, one participant said that a cancer patient program should be specially structured:

“I have run the forest healing program when the public health center was operating a support program for cancer patients as one of the subprograms such as art and music”.(FGI A-1)

“I have usually run forest healing programs for clients with stage 3 and 4 cancers. Therefore, I have accepted the offer to run the program at a nursing home. However, soon some clients were missing the class owing to being admitted to hospitals or death, which changed the dynamics of the program week to week and day to day”.(FGI-A-2)

##### Unexpected Encounter with Clients with Cancer

This theme involves the experience of forest healing instructors when they discovered that there was a cancer patient in their program. Some participants responded naturally, while others said they were embarrassed. In terms of embarrassment, there were concerns about prejudice against cancer and cancer patients and that the program might be difficult for cancer patients.

“You know… that cancer patient…, I have never known that she had cancer until she told me. I thought, “Really? She has cancer? And it’s all spread too?”.(FGI-B-5)

“I am a little concerned when others tell me about a patient having cancer. Then, I think twice about my program; when we take a walk in the forest and do some exercise, would it be appropriate for her, and what if this is too much for her, etc.?”.(FGI-C-2)

#### 3.2.2. Yearning for Healing

The cancer patients who the participants had met in the forest healing programs had a strong desire to get better and a longing for good health. With every breath, every step, and every word, the desire to heal in the forest was conveyed. Nevertheless, some cancer patients could not fully explore their passion because of their physical limitations, and it was sad to see them struggle with their limited strength.

Cancer patients with cancer who met with forest healing instructors formed a strong bond with them. The participants felt proud of seeing the results of forest healing through improvements in the health and complications of cancer patients who participated in the program for several months, and they experienced the healing power of nature once again.

##### Hope to Be Healed in the Nature

Cancer patients visit the forest for many reasons. Those who came to the forest had a great desire to be cured and freed from cancer. The desperate desire to recover the body and mind that were tired from long-term treatment was conveyed to the forest healing instructors.

“When we were sunbathing in the forest, I said to the members, ‘Think that bad energy will evaporate, and the good energy will come to you with the sunshine.’ Then she took off all the clothes, except for the underwear. She didn’t care that there were other people. I felt so sad”.(FGI B-2)

##### Know your Physical Limitations

Forest healing instructors skillfully observed members’ motions and health conditions during the program sessions. They paid close attention to cancer patients during program activities so that they participated at an appropriate level according to their physical strength.

“We were doing basic stretching, but she couldn’t do it. Later I found out she has breast cancer. Since then, I paid more attention to her. I know I can learn from books, but I don’t have any personal experiences, so for me, it was hard to understand what levels her body hormones were at and what was her physical ability and potential to do the stretching. I was depressed about it, but now I know I can let them do what they can”.(FGI-D-2)

##### As Much Time as Possible with Nature

Participants saw that the health of cancer patients who were unable to come to the forest at first, owing to physical limitations, noticeably improved with repeated visits to the forest, reconfirming the healing power of nature.

“Months have passed; someone told me she looked different before. ‘Her facial complexion has improved,’ he said. Another person told me the same. I wanted to see it myself. Therefore, I waited for her and noticed that she looked much better than before, bright, and happy. I said, ‘You look wonderful.’ She said ‘Yes, I feel wonderful. Indeed, I feel very good’”.(FGI B-4)

#### 3.2.3. People Who Need Special Care

This theme pertains to the characteristics of cancer patients encountered by participants who have run forest programs in various situations. Cancer patients have a coexistence of sensitivity and strength; therefore, they need special care. However, consideration must be given to ensure that there is no discrimination of others.

##### People Who Are Sensitive

A common characteristic of cancer patients noticed by the participants was their sensitivity. Members who wanted to hide that they were cancer patients showed sensitive reactions differing from those of the general public, such as overreacting to trivial words and actions of others.

“I need to be prepared for their sensitivity about their condition. I shouldn’t be mad about their sensitivity”.(FGI C-4)

##### People with Inner Strength

Another key characteristic of cancer patients is their inner strength. Cancer patients are people who accept the importance of daily life and health more than anyone else as an existential issue. Through them, the participants learned that appearance is not everything, and it served as an opportunity to overcome prejudice against cancer patients.

“What they have in common is that they are confident of their outcome. They think they will all recover from the surgery. Actually, their confidence is not so much about getting better physically, but it is more like they are ready to face what comes in their way”.(FGI-D-3)

##### People Who Require Special Caution when Approached

This theme of regret and difficulty emerged around participants not being able to take proper care of cancer patients because they are unaware of their emotional problems and physical symptoms. For non-medical forest healing instructors with inadequate knowledge about cancer, patients with cancer posed a great burden.

“I can’t help but feel uncomfortable. In fact, they are patients, and we are not doctors. Therefore, on the back of my mind, I am not sure what to do if things go wrong. But you know… I just do it. Honestly speaking, I am not that comfortable, especially in the beginning”.(FGI-B-1)

#### 3.2.4. Things to Prepare for Cancer Patient Programs

This theme concerned the participants’ opinions on the elements needed when planning a program for cancer patients. They learned not only from their own experiences but also from the experiences of other colleagues who participated in the FGI. They emphasized the need for individual competency in development efforts.

##### Eliminating Prejudice against Cancer Patients

Most participants came to confirm their inherent prejudice against cancer patients when running a program for cancer patients or in an unexpected situation. Everyone agreed that overcoming prejudice was the most basic preparation, and they requested opportunities, such as education, to achieve this.

“I had judged that people have cancer because they didn’t take good care of themselves regardless of whether they had a good life or not. I didn’t consider it was right of me to think that way, especially as a healing program instructor. I am now trying to understand this better…”.(FGI-D-2)

##### Specialization of Programs and Preparation to Be a Leader

There is a high demand for education regarding the appropriate level of activity for cancer patients. A medical understanding of what activities should be carefully performed and to what extent a cancer patient can perform a given activity is needed. Additionally, indoor activities have become an important factor in cancer patient programs. Participants said that they needed to strengthen their capabilities as forest healing instructors.

“I don’t know how to teach clients with breast cancer to perform arm exercises. I don’t even know what range of motions they can perform. I wish someone teaches us how to do it. It is really upsetting, as we are not medically trained, especially when the program instructors are males. It would be beneficial if there is a manual to follow, and also a training for all the healing program instructors would be nice”.(FGI D-3)

##### Spreading the Healing Power of Nature

The participants wanted to deliver the maximum healing power of nature possible to cancer patients. They said that their role was to deliver the power of nature to cancer patients by finding the best place and delivering natural healing power.

“I think twice about if the city forest is the right place for her to relax. If the city forest is the place for her, what course can she use, and would she actually feel relaxed and rested there? Where in the mountain is the best place she can find?”.(FGI4-1)

## 4. Discussion

The aim of this study was to explore the experiences of forest healing instructors in facilitating forest therapy programs for cancer patients and to understand the forest healing instructors’ perspectives on cancer patients and the forest healing program process for cancer patients through their experiences.

We derived four themes through four FGIs in which sixteen forest healing instructors participated: “prepared encounters and unexpected encounters,” “yearning for healing,” “people who need special care,” and “preparation for a cancer patient program.” Through these four topics, we found that forest therapy instructors encountered cancer patients in forest healing sites in various ways and thought that preparation was necessary to meet their health care needs.

Regarding the first theme (“prepared encounters and unexpected encounters”), cancer survivors need continuous health care because of risk of recurrence, fatigue, sleep disorders, body image damage, and job and role changes [24,25]. After being discharged from hospitals, cancer survivors seek healthy lifestyle methods to prevent recurrence and various life problems [25]. Among them, forest activities provide cancer survivors with physiologically and psychologically positive recovery experiences, and the demand for urban forest programs is high [14,26]. Therefore, cancer patients and survivors become interested in and participate in forest healing programs, and the participants of this study experienced unexpected encounters with cancer patients. In addition, they encountered cancer patients through collaboration with local integrated cancer support programs. The participants were embarrassed to meet cancer patients for the first time in the program and experienced concerns and difficulties in operating a regular, standardized program owing to the health characteristics of the cancer patients. In addition, one participant stated that the bright and lively cancer patient he encountered was different to the image he held of a typical cancer patient. This was thought to be because forest healing instructors tend to have a negative image of cancer patients that includes physical weakness and death [26]. With the latest medical advances and the development of active adjuvant therapy, it has become increasingly important to regard cancer as a chronic disease and manage a healthy lifestyle [27]. Under these circumstances, the healing factor of forests promotes recovery, even in cancer patients [13]; therefore, cancer patients’ participation in forest healing programs will increase. It is recommended that forest healing instructors should prepare for the health characteristics and emotional changes of cancer patients to guide them through the course.

The second theme was “yearning for healing.” While running the program, participants felt that cancer patients aspired to be cured of their diseases through the elements of nature. In addition, it was confirmed that when cancer patients were exposed to nature, and their health improved.

A program utilizing forest healing factors can promote physical and mental health, prevent mental disorders and physical diseases, and manage stress [28]. In addition, it helps to significantly improve quality of life [28]. As such, the healing elements of forests affect human health resilience [29]. This is supported by the attention-recovery theory which suggests that time spent in nature provides comfort, which restores attention and helps restore physiological and psychological health [30]. As cancer survivors face fear of recurrence, time spent in nature can play an active role in healthcare [24,31]. To prevent various life problems and recurrences, different methods of leading a healthy lifestyle are sought [25]. Forest healing programs improve the ability of cancer patients to control and process events that occur in their daily lives, thereby having positive impacts on stress [11]. In addition, cancer patients experience recovery through forests, which improve their psychological and physiological health, promote recovery from fatigue, reduce psychophysiological stress, and improve their positive emotional state [13]. For an efficient forest healing program, various environmental factors and characteristics of the participants should be considered [29]. Thus, it is necessary to develop a specialized forest healing program for cancer patients by applying patient-centered integrated treatments aimed at managing symptoms and optimizing health.

The third theme was “people who need special care.” Participants felt that cancer patients were sensitive and cautious and, at the same time, felt that they needed help. Cancer patients experience physical and mental difficulties owing to the side effects that occur during the treatment course [32]. Distress caused by the disease causes sensitive reactions and excessive stress throughout daily life [11]. In addition, emotional burden may result in depression, anxiety, loneliness, and stress, and they may feel uncertain about future treatment [33]. The reduction in distress in cancer patients is highly correlated with survival because it promotes physiological changes along with psychological stability [34]. Forest healing programs are the best recovery environments for reducing stress by utilizing the various environmental factors provided by forests. In addition, natural activities contribute to social adaptation by enabling cancer patients to empathize with other cancer patients and to immerse themselves physically and mentally [11]. Forest healing instructors play an important role in improving health by planning, developing, and applying customized forest healing programs to efficiently experience these effects of nature [35]. In addition, their competency and expertise are important factors in program operation [36]. In the case of cancer patients, information should be provided with education and explanations tailored to their level, taking into account their needs and preferences [37]. In order to do this, it is necessary for forest healing instructors to understand cancer patients; however, most of them are non-medical workers and may lack understanding of cancer patients. In addition, there may be a gap in the curriculum of forest healing instructor training programs that mainly deal with the general population and less with cancer patient care. Based on the current study, we hope that specific forest healing instructor trainings for forest healing programs targeted at cancer patients will be conducted in the future. In addition, it is necessary to organize and supplement forest healing programs that consider the needs and health characteristics of patients with cancer. To this end, it is important to develop programs via a multidisciplinary approach based on input from doctors, nurses, and physical therapists and to educate forest healing instructors accordingly.

The fourth theme was “things to prepare for cancer patient programs.” Within this topic, participants had difficulty in applying the program for cancer patients because they did not know about the characteristics of cancer patients. For example, it is difficult to run a forest healing program for cancer survivors owing to fear of emergencies, sensitivity, or lack of knowledge about the participants [14]. Forest healing instructors articulated the need for help from medical and health experts who are well aware of the specific needs of cancer patients. They also believed that cancer patients needed a place where they could experience the special rest and healing provided by nature.

As cancer patients show different side effects of treatment depending on the type of cancer, when providing a program such as an exercise, the method of application and considerations differ depending on the characteristics of the cancer survivor [26]. Therefore, separate locations and application methods that consider a patient’s health condition are required. In addition, smooth communication with medical staff is an important part of the treatment for cancer patients, and effective communication plays an important role in the emotional health of cancer survivors [26,32]. However, forest healing instructors may have a negative image of cancer patients and have found it difficult to communicate with them [26]. Owing to the nature of forest healing programs, they also expressed concerns about preventing emergencies and checking the health status of participants [26]. Even for forest healing instructors who are proficient in conducting forest healing programs, it may be difficult to participate in professional healing activities if they lack an understanding of the specific target group [26]. Forest healing instructors in charge of forest healing programs for cancer patients require additional competence owing to the characteristics of the patients [26]. Therefore, there is a need to seek interventions for caregivers in forest healing programs from an integrated nursing perspective for cancer patients. In addition, forest healing instructors need professional education from health and medical experts on communication with cancer patients, program operations, and emergency situations.

## 5. Limitations

This study has several limitations. Most of the participants in this study were women, and not many of the currently working forest healing instructors had experience in conducting programs for cancer patients. However, adequate data saturation was achieved by conducting interviews with multiple focus groups. It was possible to confirm the difficulties and demands of program operators by understanding the experiences of forest healing instructors running programs for cancer patients. This study provides important data for developing educational materials and guidelines for program operators.

## 6. Conclusions

This study was conducted to understand the perspective of forest healing instructors regarding their experience in running cancer patient programs. Four themes were derived from FGIs. This study confirmed the difficulties and demands of program operators working with cancer patients. To operate a forest healing program for cancer patients successfully, the program must reflect the characteristics of cancer patients, and the forest healing instructors must have the capacity and confidence to operate the program. To this end, it is necessary to continuously educate forest healing instructors on the characteristics of cancer patients to enhance their capabilities. Furthermore, administrative support is needed for the stable operation of urban forest healing center programs for cancer survivors in the community.

## Figures and Tables

**Table 1 ijerph-20-04468-t001:** General characteristics of participants (N = 16).

Characteristics		FGI 1	FGI 2	FGI 3	FGI 4	TOTAL
		N (%) or Mean (Range)				
Number of participants		5 (31.2)	4 (25.0)	3 (18.8)	4 (25.0)	16 (100.0)
Age (years)		63.20	57.00	54.67	56.75	58.44
		(57–67)	(47–63)	(52–56)	(51–61)	(47–67)
Sex	Male	3 (60.0)	1 (25.0)	0 (0.0)	0 (0.0)	4 (25.0)
	Female	2 (40.0)	3 (75.0)	3 (100.0)	4 (100.0)	12 (75.0)
Education	≤High school	0 (0.0)	0 (0.0)	0 (0.0)	1 (25.0)	1 (6.3)
	Bachelor’s degree or higher	2 (40.0)	1 (25.0)	0 (0.0)	0 (0.0)	3 (18.8)
	Graduate school	3 (60.0)	3 (75.0)	3 (100.0)	3 (75.0)	12 (75.0)
Major	Majors in forest science	1 (20.0)	2 (50.0)	0 (0.0)	3 (75.0)	6 (37.5)
	Medical, Nursing, Health Sciences	2 (40.0)	1 (25.0)	2 (66.7)	1 (25.0)	6 (37.5)
	Others	2 (40.0)	1 (25.0)	1 (33.3)	0 (0.0)	4 (25.0)
Forest welfare expert certificate	Forest healing advanced instructor	2 (40.0)	1 (25.0)	3 (100.0)	2 (50.0)	8 (50.0)
	Forest healing instructor	5 (100.0)	3 (75.0)	1 (33.3)	3 (75.0)	12 (75.0)
Career (months)		64.00	43.50	56.33	52.00	54.44
		(48–72)	(36–61)	(13–84)	(29–90)	(13–90)
Currently working as a forest healing instructor		4 (80.0)	4 (100.0)	3 (100.0)	3 (75.0)	14 (87.5)

**Table 2 ijerph-20-04468-t002:** Themes and sub-themes.

Theme	Sub-Theme
Prepared encounter and unexpected encounter	Request a special program for cancer patients
	Unexpected encounter with clients with cancer
Yearning for healing	Hope to be healed in the nature
	Know your physical limitations
	As much time as possible with nature
People who need special care	People who are sensitive
	People with inner strength
	People who require special caution when approached
Things to prepare for cancer patient programs	Eliminating prejudice against cancer patients
	Specialization of programs and preparation to be a leader
	Spreading the healing power of nature

## Data Availability

The data presented in this study are available upon request from the corresponding author.

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
