# Peer review of "Experiences of Forest Healing Instructors Who Met Cancer Patients in Forest Healing Programs: FGI Research"

_ijerph, 2023, doi:10.3390/ijerph20054468_

Round 1
Reviewer 1 Report
please review comments on the manuscript
for the discussion you added evidence about survivors which need to reconsidered as no aspect of it was in the background
this is an innovative approach that needs to be pursued with well educators
your sample was small which is ok but further exploration is recommended

Reviewer 2 Report
This is a good report on "forest healing" practiced in South Korea and the potential for care for cancer patients. However, please add additional explanations and considerations on the following points. 1. The word "forest healing" is used as a keyword in this research. What does "forest healing" refer to? Is it registered commercial law? Or is it a public, general term like horticultural therapy or music therapy? Please briefly explain this "forest healing" in this paper. 2. What kind of curriculum does a "Forest Healing" instructor complete and obtain a qualification? In the first place, isn't there a gap between the curriculum and care for cancer patients? 3. You have concluded that programs and locations that reflect the characteristics of cancer patients and education of forestry instructors are necessary, but are these necessary and sufficient conditions? Do you need anything else?
4.What are the benefits of utilizing forest spaces for patients with cancer?
Reviewer 3 Report
The author used FGI methods to address the development forest healing program for the take care of cancer patients. Author try to describe the demand of urban forest how it increase survival rate of cancer patients, by understanding the experiences of forest healing instructors who run forest healing programs for cancer patients.
I think the topic of the research is very line-up with the field, Institutional Ethics Review Board approved the study which means that the given study is original.
The author use integrated approach by using FGI and focus group interview, which provide the practical and field base analysis subject to the area of interest and objectives relevancy.
The author divide the methodology in sin different section including: 31study design (the authors provide a complete framework how the study was designed, how to set the variables to achieve the objectives and provide complete conclusions.
2-Participants and Recruitment, participate and recuriate the licensed forest healing instructor for in-depth data collection.
3 and 4-Data collection and analysis: The author use 4 FGI,S and investigate them by using qualitative and thematic analysis proposed by Baraun and Clarke and then classify them according to their nature.
5- Rigor: was applied for suitability, auditability, and verifiability criteria to ensure the accuracy of material and data nature.
6- Ethical considerations
The study was approved by Institutional Ethics Review Board (IRB: 1044396-202103-HR-055-01) which reflect the suitability of the paper for publication.
The author arguments are consistent with proposed objectives, as describe in table 1 and table 2.
Total 36 paper were cited, making the study appropriate.
Reviewer 4 Report
The abstract summarizes very clearly the purpose, methods and main results of the paper.
The Introduction is well rounded, based on relevant source from the existing literature and putting forth the relevance of this research topic.
A definition of the forest healing instructor, as used in the present research, should be provided. Some other terms such as forest interpreter, forest trekking guide, Level 1 and level 2 instructor etc. should be also defined.
It should be mentioned how the focus group interview guideline was created.
Table 1 should be restructured in order to cover less space, not 2 pages.
The 4 themes discussed in the Chapter Discussions could be presented also in more schematic, structured, visually appealing way.
Reviewer 5 Report
1. This study deals with forest healing program instructors' experience rather than participants' effects from the forest healing program. So, I would like to suggest more specific title representing this study.
2. Again, the purpose of the study should be refined and specified to argue authors' intention and appeal the importance of the study.
3. The study should include more research reviews on the effects of cancer patients' from forest/nature use or healing program; and importance/influence of healing instructors' role/experience. Those are the critically related topic with the study.
4. Still, this study didn't clearly explain how the participants of the study were selected. The study mentioned the criteria of the selection, but how the participants(FGI) were selected from population group?
5. The data were analyzed and presented by descriptive/frequency tables. Even though small number of participants, any other non-parametric analyses available?
Round 2
Reviewer 1 Report
i am perfectly fine with the changes made by the authors
great work
Reviewer 2 Report
Yes, the paper has been revised and added adequate explanation.
This paper will give a certain impact on Forest Healing in Korea!